# Farmers' Preference, Yield, and GGE-Biplot Analysis-Based Evaluation of Four Sweet Potato (*Ipomoea batatas* L.) Varieties Grown in Multiple Environments

Abdullah Al Mahmud [1], Mohamed M. Hassan [2,*], Md Jahangir Alam [1], Md Samim Hossain Molla [3], Md Akkas Ali [4], Haridas Chandra Mohanta [5], Md Shahidul Alam [6], Md Aminul Islam [6], Md Alamin Hossain Talukder [3], Md Zannatul Ferdous [3], Md Ruhul Amin [7], Md Faruque Hossain [4], Md Mazharul Anwar [8], Md Shahidul Islam [9], Eldessoky S. Dessoky [2] and Akbar Hossain [10,*]

1    On-Farm Research Division (OFRD), Bangladesh Agricultural Research Institute (BARI), Gaibandha 5700, Bangladesh; mahmud.tcrc@gmail.com (A.A.M.); jahangir.bari@gmail.com (M.J.A.)
2    Department of Biology, College of Science, Taif University, P.O. Box 11099, Taif 21944, Saudi Arabia; es.dessouky@tu.edu.sa
3    OFRD, BARI, Rangpur 5400, Bangladesh; samimmolla@gmail.com (M.S.H.M.); alamintalukder@yahoo.com (M.A.H.T.); zferdous80@gmail.com (M.Z.F.)
4    OFRD, BARI, Joydebpur 1701, Bangladesh; akkasbari@gmail.com (M.A.A.); faruque1969@yahoo.com (M.F.H.)
5    Regional Horticulture Research Centre, BARI, Chapainawabganj 6300, Bangladesh; hmohantatcrc@gmail.com
6    OFRD, BARI, Bogra 5800, Bangladesh; alamsrc84@yahoo.com (M.S.A.); amin.agron@gmail.com (M.A.I.)
7    OFRD, BARI, Manikganj 1801, Bangladesh; aminagr70@gmail.com
8    OFRD, BARI, Rajshahi 6201, Bangladesh; anwar.sci.bari@gmail.com
9    OFRD, BARI, Bhola 8300, Bangladesh; shahid75bari@gmail.com
10   Department of Agronomy, Bangladesh Wheat and Maize Research Institute, Dinajpur 5200, Bangladesh
*    Correspondence: m.khyate@tu.edu.sa (M.M.H.); akbarhossainwrc@gmail.com (A.H.)

**Abstract:** The study aimed to select high-yielding, farmers-preferred quality sweet potato varieties that are suitable to grow in various environmental conditions in Bangladesh. In this context, four popular sweet potato varieties (viz., 'BARI Mistialu-8', 'BARI Mistialu-12', 'BARI Mistialu-14', and 'BARI Mistialu-15') were used in the study. These varieties were released by Bangladesh Agricultural Research Institute (BARI). In the first season (2018–2019), these varieties were evaluated at nine locations, and in the second season (2019–2020), the same varieties were tested further, but only in three locations. The trial was set up in a randomized complete block design and repeated three times. After two years of observation, it was found that the fresh root yield was varied significantly due to the environment (E), genotypes (G), and their (G × E) interaction ($p \leq 0.01$) by using genotype and genotype x environment (GGE) biplot analysis. The E and G × E interaction effects were found to the greater than the genotypes effect solely. In the first year, three varieties, namely 'BARI Mistialu-8', 'BARI Mistialu-12' and 'BARI Mistialu-14', were identified as balanced and comparatively higher in yield in nine locations. These three varieties also showed a similar trend with respect to root yield in tested three locations in the second year. Among the four varieties, 'BARI Mistialu-12' was found to be the highest root yielder, followed by 'BARI Mistialu-8' and 'BARI Mistialu-14'. Across the locations, these varieties showed 57.89%, 61.50% and 44.30% higher yield than the local check cultivar. Therefore, these three varieties may be recommended as the best varieties of sweet potato throughout the country.

**Keywords:** sweet potato; yield; quality; smallholders; GGE biplot analysis; on-farm trial

## 1. Introduction

Sweet potato (*Ipomoea batatas* L.) is rich in many vitamins, minerals, and beneficial fibers [1]. The unpretentious sweet potato's antioxidant, vitamin, and mineral values make it a "superfood". Calorie-wise, sweet potato is an ideal food [2]. Naturally sweet, one

medium potato has only 105 calories with four grams of fiber and, unless it is served with butter, zero fat [3,4]. It also supplies 438% of the daily value of vitamin A and 37% of the daily value of vitamin C, as well as being a good source of important B vitamins, manganese, copper, and iron [5]. There is also good evidence from medical studies that antioxidants in sweet potatoes may be beneficial in preventing several chronic and deadly diseases, including diabetes and cancer [2,6]. At present in Bangladesh, children in rural and char areas are highly vulnerable to night blindness caused by vitamin A deficiency, which affects 2% of all children of 1–6 years of age; each day, about 88 children become blind [7]. This can be partly ameliorated by nutritional intervention by way of education to promote awareness of the benefit of eating orange-fleshed sweet potato (a rich source of vitamin A). Generally, poorer people in char lands (riverbanks) are growers and consumers of sweet potato, which is commonly known as Mistialu in South Asia, particularly in India, Bangladesh, Sri Lanka, and the Maldives.

The total production of sweet potato in Bangladesh increased 3.16% (from 254,633 to 262,702 metric tons (MT) in 2015 to 2017), and area increased by 1.94% (from 25,260 to 25,750 ha in 2015 to 2017) [8,9]. This is due to the introduction and adoption of Bangladesh Agricultural Research Institute (BARI) releasing modern varieties, improved cultivation techniques, and awareness building by sweet potato growers. This crop also contributes greatly to farmers' income in Bangladesh [10]. The typical yield of sweet potato in Bangladesh is only 10.20 t ha$^{-1}$ [9], while the probable or achievable yield has been stated to be as much as 40 t ha$^{-1}$. There are considerable opportunities for increasing the yield of sweet potato by limiting the yield gap [10]. With escalating demand for more food to meet the demands of an ever-increasing population, it is essential to explore the possibilities of Bangladesh's vast lands for the heightened production of sweet potatoes.

Most rural families in developing countries are directly dependent on principal crops and farming for food. All over the world, sweet potato is recognized as a food security crop for its numerous resilient features. The key traits of sweet potato's growth success encompass its capacity to grow on both rich and poor sandy soils, its potential to grow non-seasonally in tropical regions, its higher root yield ha$^{-1}$, its drought and salt tolerance, and its resistance against some pests and diseases. These characteristics allow sweet potato plants to make a return while feeding people. According to Chueyen and Eun [11], sweet potato is the seventh most consumed carbohydrate-rich food in the world. China is the world's major sweet potato producer and produces around 71 million tons per year. Sweet potato delivers the highest comestible energy (ha$^{-1}$ day$^{-1}$) of all native food crops in sub-Saharan areas. Sweet potato also is a key crop in Burundi, Malawi, Rwanda, and parts of Uganda [12].

However, the yields of sweet potato are vulnerable to the genetic makeup of cultivars, environmental conditions, and their interaction, as uncovered by Wolfgang et al. [13], Chiona [14], Osiru et al. [15], and Moussa et al. [16]. Orange-fleshed sweet potato (OFSP) varieties have been evaluated in the southern part of Bangladesh, and it was found that most OFSP clones were responsive to environmental differences [17]. Wolfgang et al. [13] carried out an experiment on genotype (G) × environment (E) interactions for a diverse set of sweet potato genotypes across various eco-geographic situations in Peru and described an important G × E interaction of a cross-over nature. This may suggest the existence of disparity of sweet potato genotypes crossways locations and years. The genotype and genotype × environment (GGE) biplot display both genotype (G) and genotype × environment interactions (GEI), which are the two sources of variation that are relevant for genotype evaluation [18]. The GGE biplot is constructed by plotting the first two principal component axis (PCA1 and PCA 2) developed from singular value decomposition (SVD) of the environment-centered data. Simulations that decompose the environment-centered data are usually mentioned as sites regression models or SREG, and SREG with two PCs is referred to as SREG2 [19]. The GGE biplot which has significance connecting with two major aspects including PCA1 and PCA2 are one of them shows pattern of environmental data and helps find high yielding and stable cultivars. Second of them

is used deciding the capability and representativeness of the experiment environments extracted from SREG2 model [19]. It contains useful information about genotype yield and stability effectiveness. Furthermore, it can pinpoint environments with the power to differentiate between genotypes and to assess the representativeness or stability of the destination environments [18,20]. Interaction of genotype plus genotype by environment (GGE biplot) were advanced by Yan et al. [19], Yan et al. [21], Yan [22], Hossain et al. [23].

In Bangladesh, there is an inadequate report on the G × E interactions and the constancy of capable sweet potato genotypes. Therefore, identifying the nature of G × E interaction, measuring the scale and finding steady and broadly adapted sweet potato varieties/genotypes is necessary before release and large-scale spreading. Documentation of the utmost selective and illustrative trial locations is important for additional variety valuation and cultivation of sweet potato in Bangladesh.

There are many instances in the world where varieties were pinpointed and released with better qualities that were not attractive to the farmers or the users. Researchers also tend to watch the traits that are of concern to the growers and users. This has led to the low rate of spreading after the release of the varieties. The main obstacle with this approach was the lack of participation of the stakeholders like the growers and consumers in evaluating and selecting the varieties. The participatory varietal selection (PVS) method has been fruitful in pinpointing the varieties in a shorter time by involving growers and consumers. It has also led to faster spreading and boosted cultivar diversity. Moreover, it is also stated that by following the PVS method, research costs can be decreased, and the adoption rate can be expanded. This method considers the preferences of the growers and their socio-economic positions when evaluating and choosing varieties [24].

In recent years, Bangladesh Agricultural Research Institute (BARI) has developed several high-yield beta-carotene-enriched sweet potato varieties that are able to grow in unfavorable situations like drought and salinity to satisfy the daily intake of vitamin A. For large-scale dissemination throughout the country, these varieties need on-farm validation trials in the various agro-ecological zones (AEZ) to identify the appropriateness of the different varieties and get feedback from the farmers.

## 2. Materials and Methods

### 2.1. Experimental Locations

The study was carried out in nine districts i.e., Rangpur (E1), Gaibandha (E2), Bhola (E3), Jhenaidah (E4), Manikganj (E5), Kishoreganj (E6), Sylhet (E7), Norsingdi (E8) and Bogura (E9), representing nine different agro-ecological zones (AEZ) of Bangladesh [25] (Table 1) during the winter seasons of 2018–2019 and 2019–2020. Meteorological data on total rainfall and the minimum and maximum temperature in sweet potato growing season is presented in Figures 1 and 2. The monthly average rainfall pattern is quite similar across the locations during the whole crop growing period (November to May), with some exceptions in Sylhet (E7). In Sylhet, the crop received more rainfall (average 169 mm) in the whole growing period, with the maximum (375.6 mm) in April. At all the locations, the crop received more rainfall after February (at the vegetative stage) and continued until April, which helped better crop growth and development. The temperature slightly went down from November to January in a declining trend and after that again increased gradually at all the locations of the experimental areas. The monthly average maximum temperature ranges from 21.8 to 36.3 °C across the locations, whereas the lowest value (21.8 °C) is in Gaibandha (the northern part of Bangladesh) (E2) in January and the highest temperature (36.3 °C) is in Jhenaidah (E4) in April. Similarly, the monthly average minimum temperature ranges from 10.7 to 27 °C across the different agroecological zones, with the minimum (10.7 °C) in Rangpur (E1) in January and maximum (27 °C) in Manikgonj (E5) in April. All the weather data are the average value collected from Bangladesh Metrological Department.

**Table 1.** Soil characters of nine experimental sites representing their agroecological zones (Information source: BAMIS [25].

| Location | Representing Agro-Ecological Zone (AEZ) | Major Soil Characters of Agro-Ecological Zone (AEZ) |
|---|---|---|
| E1 (Rangpur) and E2 (Gaibandha) | AEZ 2: Active Tista Floodplain | Irregular patterns of grey stratified sands and silts, very strongly acidic to slightly acidic (pH range: 3.8 to 6.4). Very low to low organic matter content. Zn and B content with soil fertility level is very low to medium. |
| E3 (Bogura) | AEZ 4: Karatoa-Bangali Floodplain | Soils are grey silt loam to silty clay loams on ridges and dark grey clays in basins Soils are strongly acidic to slightly acidic (pH range: 4.1 to 6.5. Organic matter content is low to medium. |
| E4 (Jhenidah) | AEZ 11: High Ganges River Floodplain | Common soil types are calcareous brown floodplain soil, silty loam to silty clay loam, acidic to alkaline in nature (pH range: 4.5 to 8.3) with low to medium organic matter content. The general fertility level is low, including N, P, S, B and Zn, with medium to high K-bearing minerals. |
| E5 (Manikganj) | AEZ 8: Young Brahmaputra and Jamuna Floodplain | Soils are characterized by silty loam to silty clay loam on the ridges and clays in the basins; slightly acidic to neutral (pH range: 4.5 to 7.2). Soils are deficient in N, P, S, B, K and Zn, with low to medium organic matter content. |
| E6 (Kishoreganj) | AEZ 21: Sylhet Basin | In the higher parts, soils are silty clay loam and clay in the wet basins, with low to medium organic matter, slightly acidic (pH range: 4.6 to 6.1) in nature. Fertility level is medium to high with extremely low N and low to medium P content. |
| E7 (Sylhet) | AEZ 20: Eastern Surma Kushiyara Floodplain | Common soil types are non-calcareous grey floodplain, silty clay loams on the ridges and clays in the basins. pH ranges from 3.8 to 7.7. Organic matter content is low to medium. Soil is deficient in N, P, B and Zn. |
| E8 (Narsingdi) | AEZ 9: Old Brahmaputra Floodplain | Soils are mainly silt loams to silt clay loams on the ridges and clay in the basins. Top soils are strongly acidic to neutral, and sub-soils are neutral in reaction (pH ranges from 3.8 to 7.2). Organic matter content is low on the ridges and moderate in the basins, with low fertility status of N, P, K, S and B. |
| E9 (Bhola) | AEZ 18: Young Meghna Estuarine Floodplain | The major soils are calcareous silt loam to silt clay loams, which become saline in the dry period. Top-soils and subsoils are mildly alkaline (pH ranges from 4.3 to 8.4). Soil fertility level is low to medium with very low N content. |

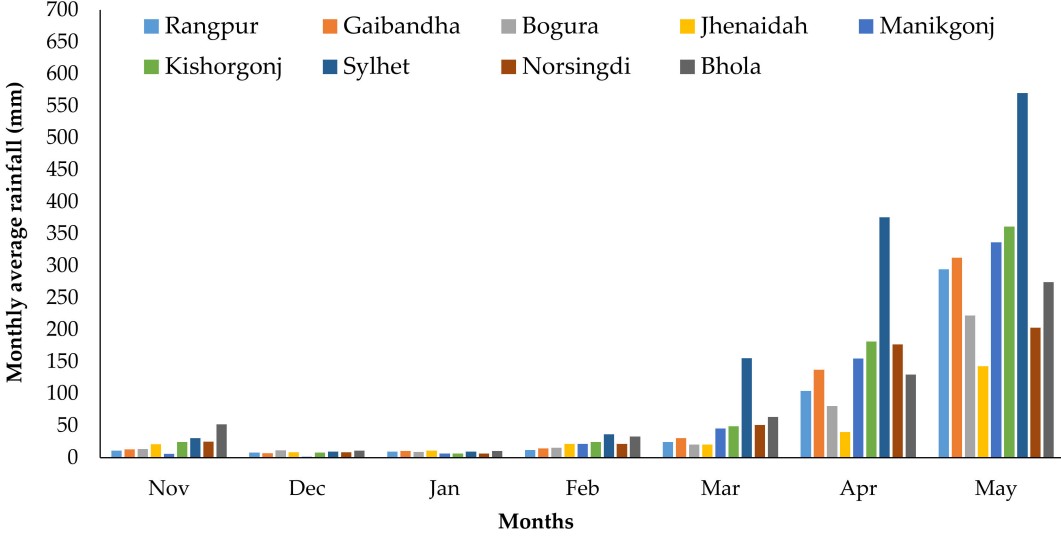

**Figure 1.** Monthly mean rainfall (mm) of the experimental sites during sweet potato growing season (average of 2018–2019 and 2019–2020 crop season).

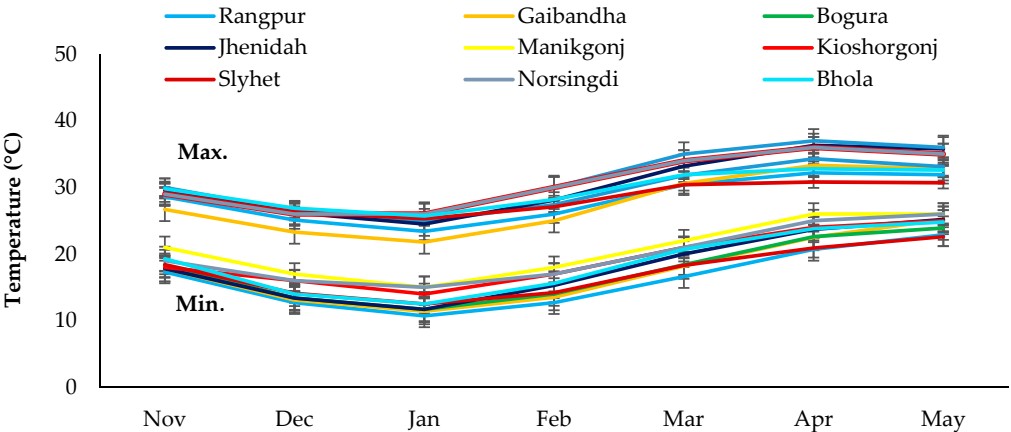

**Figure 2.** Monthly mean maximum and minimum temperature (°C) of the experimental sites during sweet potato growing season (average of 2018–2019 and 2019–2020 crop season).

### 2.2. Experimental Design and Plant Materials

The experimental materials were four BARI-released, Vitamin-A-enriched sweet potato varieties, namely 'BARI Mistialu-8' (G1), 'BARI Mistialu-12' (G2), 'BARI Mistialu-14' (G3) and 'BARI Mistialu-15' (G4) (Table 2). The trial was designed in a randomized complete block (RCB) with three dispersed replications in each of the locations. In the first season (2018–2019), the planting of vines was started on 25 October and continued until 26 November 2018 (Table 3). Unit plot size varied from location to location (Table 3).

**Table 2.** Characters of Sweet potato varieties utilized in the trial.

| Name of the Variety | Pedigree | Year of Release | Major Characters | Image |
|---|---|---|---|---|
| G1: (BARI Mistialu-8) | CIP-440025 | 2008 | Skin color: Red<br>Flesh color: Yellow<br>Dry matter: 33.71 ± 1%<br>Beta-carotene: 1.08 mg/100 g FW<br>Fe: 7.86 mg/kg<br>Zn: 14.76 mg/kg | |
| G2: (BARI Mistialu-12) | CIP-440001 | 2013 | Skin color: Yellow<br>Flesh color: Orange<br>Dry matter: 22.04 ± 1%<br>Beta-carotene: 3.60 mg/100 g FW<br>Fe: 14.76 mg/kg<br>Zn: 8.09 mg/kg | |
| G3: (BARI Mistialu-14) | CIP-441132 | 2017 | Skin color: Light orange<br>Flesh color: Orange<br>Dry matter: 29.46 ± 1%<br>Beta-carotene: 10.10 mg/100 g FW<br>Fe: 5.17 mg/kg<br>Zn: 6.47 mg/kg | |
| G4: (BARI Mistialu-15) | CIP-440267.2 | 2017 | Skin color: Pink<br>Flesh color: Orange<br>Dry matter: 28.91 ± 1%<br>Beta-carotene: 10.39 mg/100 g FW<br>Fe: 13.25 mg/kg<br>Zn: 6.47 mg/kg | |

FW, fresh weight. (Source information: Quality and nutrition Laboratory, International Potato Center (CIP), Lima, Peru).

**Table 3.** Crop management practices of adaptive trials on sweet potato at different locations.

| Crop Season | MLT Sites | Plot Size | Date of Planting | Date of Harvesting | Duration |
|---|---|---|---|---|---|
| | Pirganj–Rangpur | 10 m × 10 m | 25–26 October 2018 | 15–16 February 2019 | 113 |
| | Saghata–Gaibandha | 10 m × 10 m | 25–26 October 2018 | 14–16 March 2019 | 140 |
| | Sariakandi–Bogura | 10 m × 10 m | 17 November 2018 | 23 March 2019 | 124 |
| | Kaliganj–Jhenidah | 10 m × 10 m | 12 November 2018 | 28 March 2019 | 136 |
| 2018–2019 | Manikganj Sadar | 10 m × 10 m | 4 December 2018 | 04–05 April 2019 | 121 |
| | Kishoreganj Sadar | 10 m × 10 m | 19 November 2018 | 25 March 2019 | 126 |
| | South Surma, Sylhet | 10 m × 10 m | 24–26 November 2018 | 08–09 April 2019 | 135 |
| | Shibpur–Norsingdi | 5 m × 6 m | 19 November 2018 | 15 April 2019 | 147 |
| | Daulatkhan–Bhola | 6 m × 6 m | 11 December 2018 | 06 May 2019 | 146 |
| | Saghata–Gaibandha | 40 m × 30 m | 26 October 2019 | 10–12 March 2020 | 135 |
| 2019–2020 | Kaliganj–Jhenidah | 40 m × 30 m | 12 November 2019 | 25 March 2020 | 133 |
| | Shibpur–Norsingdi | 40 m × 30 m | 15 November 2019 | 2 April 2020 | 138 |

MLT, Multi-location trial.

Based on the production root biomass in the 1st year (2018–2019) and G × E interaction, three selected sweet potato varieties were tested in the 2nd year (2019–2020) in three different locations of Gaibandha, Jhinaidah and Norsingdi, but only in progressive farmers' fields and replicated thrice in each location. In the second year, planting of the sweet potato vines was commenced from 26 October and continued until 15 November 2019 across the locations. The plot size, planting and harvesting time with crop durations are presented in detail in Table 3. Root yield of the selected sweet potato varieties was compared with the local variety.

*2.3. Experimental Procedures*

The crop was planted with a spacing of 60 cm × 30 cm. Manures and fertilizers were used at a rate of 10 t ha$^{-1}$ of well-composted cow-dung and 105-45-105-15-2-1 kg ha$^{-1}$ of N-P-K-S-Zn-B, respectively, in the form of urea, TSP, MoP, gypsum, zinc sulfate and boric acid. Fifty percent of N, K and a full dose of other fertilizers were applied while closing land was being prepared. The rest amount of the N and K fertilizer was applied 35 days after planting (DAP). Crop fields were irrigated 6 times at 30, 45, 60, 75, 90, 115 DAP, maintaining 2/3rd of the valley. Sweet potato weevil was controlled by the combined approach, earthing up (30, 60 and 90 DAP) with Sex pheromone trap and, carbofuran (Furadan 5G®, Padma Oil Company Limited, Bangladesh) @ 500 g a.i. ha$^{-1}$ at 60 DAP. Harvesting of the sweet potato started on 15 February 2019 and continued until 6 May 2019. The tuberous root yield was collected from an area of 4 m$^2$ (2 m × 2 m) in each of the locations and converted into t ha$^{-1}$.

*2.4. Farmers-Preference-Based Variety Selections*

Sweet potato varieties were assessed in the harvest stage following the participatory variety selection method [17]. For the organoleptic evaluation, three panels consisted of five men, five women and five research personnel for testing of the sweet potato varieties grown in the different locations. One kilogram of sweet potatoes from each variety was boiled, and the clones were separated on different plates and identified by recording numbers. The basic rules of the assessment were clarified by evaluating a few words of the representatives of the panels. An evaluation form was given to each panel member, which was used to keep the assessment about the appearance, taste, texture and fiber of each variety. Each panel member evaluated variety by variety and washed his/her mouth with mineral water before moving on to the next sample. The appearance of sweet potato varieties was recorded based on the boiled sweet potatoes offered on plates on a 1–5 scale (where 5 = Very good, 4 = Good, 3 = Fair, 2 = Poor and 1 = Very Poor); flesh color was measured after a cross-section of each boiled sweet potato on a 1-5 scale; (where 5 = Very good, 4 = Good, 3 = Fair, 2 = Poor and 1 = Very Poor); taste was indicated based on the personal criterion on a 1-5 scale (where 5 = Very good, 4 = Good, 3 = Fair, 2 = Poor and 1

= Very poor); texture was measured on dry matter content in sweet potato on a 1–5 scale (where 5 = Mealy/Floury, 4 = Less floury, 3 = Fair/Intermediate, 2 = Soggy and 1 = More soggy/watery) and fiber content was determined from the amount of fiber present in the flesh of boiled sweet potato in 1–5 scale (where 5 = No fiber presence, 4 = Less fiber presence, 3 = Fair/moderate fiber presence, 2 = Poor/high presence of fiber and 1 = Roots are fibrous).

### 2.5. Statistical Analysis

Tuberous root yield of four sweet potato varieties under nine environmental conditions was used to combine analysis of variance (ANOVA) to ascertain the consequences of environment (E), genotype (G) and how they interact. The data were graphics-intensive studied for an interpretation of G and E and their interaction by using the R software [26]. This is called GGE-biplot methodology. The methodology of GGE-biplot comprises double perceptions, the biplot [27] and the GGE perception [19], which had been used to visually assess the sweet potato genotypes MET data. By using a biplot, this methodology demonstrated the influences (G and GE), which are significant in genotype assessment and that are the bases of variation in GE interaction analysis of MET data [28,29]. The diagrams were produced depending on (i) the polygon view of GGE biplot to the identification of winning genotypes and their mega-environments by "which-won-where" pattern, (ii) ranking of genotypes depended on yield and stability performance, (iii) assessment of genotypes associated with perfect genotypes, (iv) assessment of environments associated with perfect environments, (v) affiliation among environments and (vi) contrast between two genotypes. In the 2nd year trial, tuberous root yield, % weevil infestation, infested tuber yield and healthy root yield data of the four sweet potato varieties under three environmental conditions were analyzed and mean separation was made.

## 3. Results and Discussion

### 3.1. Combined Analysis of Variance

Sweet potato varieties grown in different agro-ecology conditions showed a considerable fluctuation in tuberous root yield (Table 4). In the case of tuberous root yield, genotype (G), environment (E) and genotype × environment interaction (GEI) was very much significant ($p < 0.001$). The very much significant G × E impacts suggest that genotypes may turn out to be cautiously chosen for an adjustment to specific environments, which is in accord with the results of Aina et al. [30] and Xu et al. [31] in G × E interaction effects of Cassava genotypes, Gurmu [29] in sweet potato genotypes. From the findings of the GGE biplot, it was found that, among the sum of squares, 91.68% was substantiated by the first two principal components, along with 59.26 and 32.42% in Axis1 and Axis2, respectively, of the GGE sum of squares. G plus GE of an MET in a manner that facilitates visual cultivar assessment and mega-environment identification was graphically shown by GGE biplot [21]. Therefore, the G × E interaction impacts expressed that genotypes reacted in a different way to the difference in ecological environments of locations, which suggested the requirement of checking sweet potato genotypes/varieties in different locations. This too reveals the complexities confronting the plant breeders in choosing new sweet potato varieties to be released. The reasons clarified (%) proposed that the tuberous root yield was influenced by genotype (76.51%), environment (12.49%) and their interaction (10.21%).

**Table 4.** Combined analysis of variance of tuberous root yield of four sweet potato genotypes assessed at nine environments in the 2018–2019 crop season.

| SOV | DF | SS | MS | PORCENT | PORCENAC |
|---|---|---|---|---|---|
| ENV | 8 | 365.0834 | 45.63542 | 57.25402 | 57.25402 |
| GEN | 3 | 105.9319 | 35.31063 | 16.61272 | 73.86674 |
| ENV*GEN | 24 | 166.6401 | 6.94334 | 26.13326 | 100 |

### 3.2. Average Yield Performance of the Genotypes in Multiple Environments

In both years, yield of all varieties were significantly varied due to the location specific environment (Figure 3 and Table 5). In the first season (2018–2019), the environment of Bhola was found the most suitable for the production tuberous root (average yield of 31.17 t ha$^{-1}$) followed by Sylhet (30.63 t ha$^{-1}$) and Norsingdi (29.59 t ha$^{-1}$). While the environmental condition of Manikganj was recorded the least productive environment (19.55 t ha$^{-1}$), followed by Bogura (26.21 t ha$^{-1}$). Among these genotype, 'BARI Mistialu-12' was the average highest root yielder (30.70 t ha$^{-1}$), and produced more than the average yield in three locations. The genotype 'BARI Mistialu-8' was the average 2nd highest root yielder (26.79 t ha$^{-1}$), and produced more than the average yield in seven locations. The genotype 'BARI Mistialu-14' was the average 3rd highest root yielder (26.67 t ha$^{-1}$), and produced more than the average yield in five locations. While, the genotype 'BARI Mistialu-15' was the average lowest root yielder (24.73 t ha$^{-1}$), and produced more than the average yield in six locations.

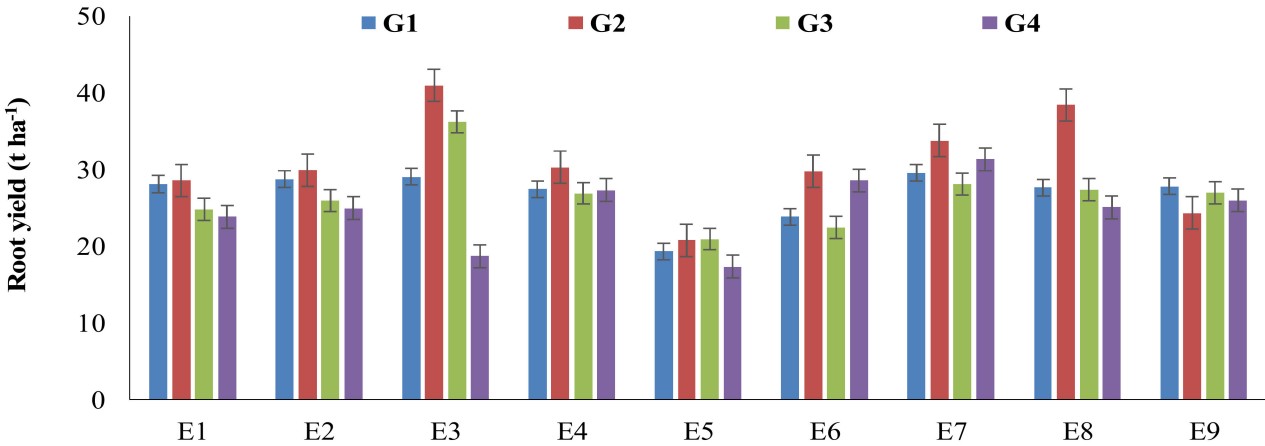

**Figure 3.** Yield performance of four sweet potato varieties in nine different locations in 2018-2019 (first year) crops season (mean ± standard error of the three replications).

**Table 5.** Root yield of selected sweet potato genotypes at farmers' fields in three environments during the 2019–2020 (second year) crop season.

| Variety | Yield (t ha$^{-1}$) | | | | % Yield Increases Over Check Variety |
|---|---|---|---|---|---|
| | Gaibandha | Norsingdi | Jhenaidah | Variety Mean | |
| G1 (BARI Mistialu-8) | 29.20 abB | 35.54 bA | 23.96 aC | 29.57 | 57.89 |
| G2 (BARI Mistialu-12) | 30.56 aB | 37.67 aA | 22.50 abC | 30.24 | 61.50 |
| G3 (BARI Mistialu-14) | 28.50 bB | 30.85 cA | 21.72 bC | 27.02 | 44.30 |
| Local cultivar (Check) | 22.30 cA | 18.53 dB | 15.35 cC | 18.73 | - |
| Location mean | 27.64 | 30.65 | 20.88 | | |

Means with the same letter are not significantly different. The small letter denotes the variations among the genotypes within the location, and the capital letter denotes the variation among the locations within the variety.

In the 2nd year (2019–2020), genotypes 'BARI Mistialu-8', 'BARI Mistialu-12', 'BARI Mistialu-14' with a local check variety were tested in three locations through on-farm validation trials (Table 5). Hence, a significant variation was found for fresh root yield among the genotypes at all three locations which are ranged from 15.34 to 30.84 t ha$^{-1}$. Among these varieties, the highest root yield was obtained in Norsingdi from 'BARI Mistialu-12' (37.67 t ha$^{-1}$); in Gaibandha also from 'BARI Mistialu-12' (30.56 t ha$^{-1}$), and in Jhenaidah from 'BARI Mistialu-8' (23.96 t ha$^{-1}$). As compared to local variety, BARI Mistialu-12, BARI Mistialu-8 and BARI Mistialu-14 produced 61.50, 57.89 and 44.30% higher yield, respectively. Considering all three locations, 'BARI Mistialu-12' (30.24 t ha$^{-1}$) was the highest root yielder, followed by 'BARI Mistialu-8' (29.57 t ha$^{-1}$).

Infested (%) and non-infested root yield due to weevil in both crop seasons were presented in Table 6, whereas infested root yield was found significant only for the 1st year (2018–2019). Weevil infestation (%) of sweet potato varieties ranged from 3.43 to 5.43% in the 1st year (2018–2019) and 3.11 to 5.80% in the 2nd year (2019–2020) crop season. In both years, low weevil infestation was observed in the variety 'BARI Mistialu-8'. Similarly, the minimum weevil-infested root yield was recorded for 'BARI Mistialu-8, which ranged from 0.90 to 1.30 t ha$^{-1}$ in the 1st year (2018-19) and 0.92 to 1.31 t ha$^{-1}$ in the 2nd year (2019-20). The mean non-infested root yield among the genotypes of all locations ranged from 22.64 to 29.46 t ha$^{-1}$ and 17.64 to 28.94 t ha$^{-1}$ during 1st year (2018-2019) and 2nd year (2019-2020), respectively. Among the varieties, 'BARI Mistialu-12' (29.46 t ha$^{-1}$) was the highest mean non-infested root yielder followed by 'BARI Mistialu-8' (25.38 t ha$^{-1}$) in the 1st year (2018–2019). In the 2nd year (2019-2020), 'BARI Mistialu-12' (28.94 t ha$^{-1}$) was the highest non-infested root yielder, followed by 'BARI Mistialu-8' (28.65 t ha$^{-1}$). The results of the study indicated that no genotypes/varieties were found resistant to root weevil. However, the average infestation was <5%, which is associated with plant traits and could be manipulated by cultural practices.

**Table 6.** % Weevil infestation (by weight mean of all locations), infested and non-infested root yield of sweet potato genotypes in both seasons.

| Variety | % Weevil Infestation (by Weight) | | Infested Root Yield (t ha$^{-1}$) | | Non-Infested Root Yield (t ha$^{-1}$) | |
| --- | --- | --- | --- | --- | --- | --- |
| | 2018–2019 | 2019–2020 | 2018–2019 | 2019–2020 | 2018–2019 | 2019–2020 |
| G1 (BARI Mistialu-8) | 3.43 b | 3.11 c | 0.90 b | 0.92 | 25.38 b | 28.65 a |
| G2 (BARI Mistialu-12) | 4.67 ab | 4.31 b | 1.44 a | 1.30 | 29.46 a | 28.94 a |
| G3 (BARI Mistialu-14) | 5.31 a | 4.84 ab | 1.38 a | 1.31 | 24.58 bc | 25.71 b |
| G4 (BARI Mistialu-15) | 5.43 a | - | 1.30 a | - | 22.64 c | - |
| Local cultivar (Check) | - | 5.80 c | - | 1.09 | - | 17.64 c |
| Mean | 4.71 | 4.52 | 1.26 | 1.15 | 25.51 | 25.24 |

Means with the same letter are not significantly different but different letter denotes the variations among the genotypes.

The high yield of the sweet potato varieties with the evaluated record was preferred by the growers. In both years, the Norsingdi location had the best suitable area with higher root yield, the reason being that higher field days allowed for root production. The overall yield performance of all studied varieties was comparatively low in Manikganj and Bogura considering their potential yield. The probable causes for its low yield due to inadequate management practices (like irrigation and weeding) applied by the farmers at the time of root initiation roots, edaphic and climatic condition and as a new crop in that area. Incorporation of vitamin-A-enriched sweet potato varieties in different agroecological zones will enrich the diversity of sweet potatoes from which selection will take place. These results support the findings of Rafique et al. [32] and Rahaman et al. [17].

Nevertheless, recommendations depending on average yield performance only might be confusing if constancy across situations is not studied. A genotype with average yields across the environments might provide maximum yield across limited environments and to a lesser extent in other environments. Conversely, there could be genotypes that are steadily accomplished across situations irrespective of ecological geographies, which may influence their performance. Therefore, constant study facilitates the detection of such kinds of genotypes for the approval for extensive adaptation.

*3.3. Organoleptic Evaluation*

An organoleptic evaluation was performed with two panels after harvest. Five male and five female farmers participated in the organoleptic evaluation of the sweet potato varieties. Considering the mean appearance, flesh color, taste, fiber and flesh texture of each genotype/variety G3 ('BARI Mistialu-14') ranked first followed by G1 ('BARI Mistialu-8') and G4 ('BARI Mistialu-15'); on the other hand, the farmer's choice was the poorest for

G2 ('BARI Mistialu-12') (Figure 4). Orange-fleshed sweet potato (biofortified crop) has an impression on Vitamin-A intake, that can be helpful to improve the health status of vulnerable people if they consume in sufficient quantities. The difference in preference for the varieties on flesh color, taste, texture and fiber stress the importance of doing organoleptic evaluation study particularly to increase the acceptability of varieties and to disseminate the selected varieties to local preferences where possible. These scores back the results of Rahaman et al. [17]; BARI Annual Report, 2014–2016 [33] and Islam et al. [34].

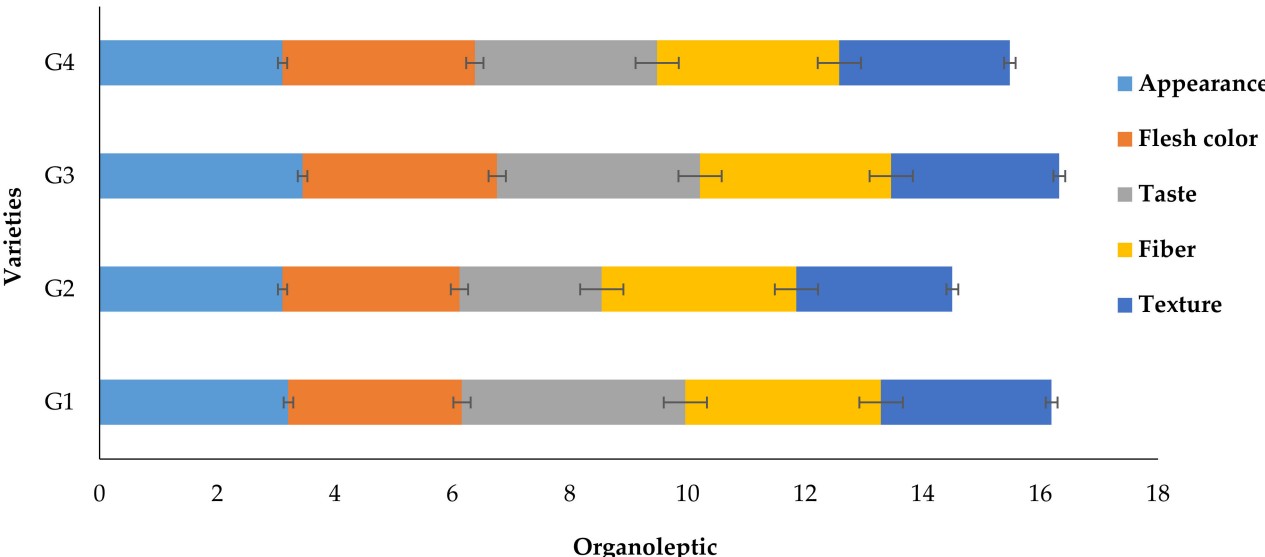

**Figure 4.** Organoleptic evaluation of four sweet potato varieties (mean ± standard error of the fifteen replications). Appearance, flesh color, taste scale 1–5 (where 5 = Very good, 4 = Good, 3 = Fair, 2 = Poor and 1 = Very Poor); texture scale 1–5 (where 5 = Mealy/Floury, 4 = Less floury, 3 = Fair/Intermediate, 2 = Soggy and 1 = More soggy/watery) and fiber Scale 1–5 (Where, 5 = No fiber presence, 4 = Less fiber presence, 3 = Fair/moderate fiber presence, 2 = Poor/high presence of fiber and 1= Roots are fibrous).

### 3.4. Winning Genotype and Mega-Environment

For analyzing the feasible existence of crop varieties in multiple places in a region, the pattern image of "which-won-where" of multi-environment trial (MET) data is essential [19,21] (Figure 5). The polygon view of a biplot is the perfect way to envision the interplay patterns among genotypes and environments and to efficiently infer a biplot [28]. The G1, G3 and G4 are the vertex genotypes in this study. The vertex genotype within the sector was provided with the highest yield for the environments. Another significant feature of Figures 1 and 2 is environmental groupings, which indicate the potential presence of various mega-environments. Therefore, depending on the biplot analysis data of nine environments, three mega-environments are proposed in Figure 5. The first mega-environment encompasses the winner environments of Gaibandha (E2), Bogura (E9), Manikganj (E5), Norsingdi and Bhola with genotype G1 ('BARI Mistialu-8') and G3 ('BARI Mistialu-14'); the second mega-environment encompasses the winner environments of E4 (Jhenaidah), E7 (Sylhet) and E6 (Kishoreganj), with genotype G4 ('BARI Mistialu-15'). The environment of E1 (Rangpur) with the winner G2 ('BARI Mistialu-12') makes up another mega-environment.

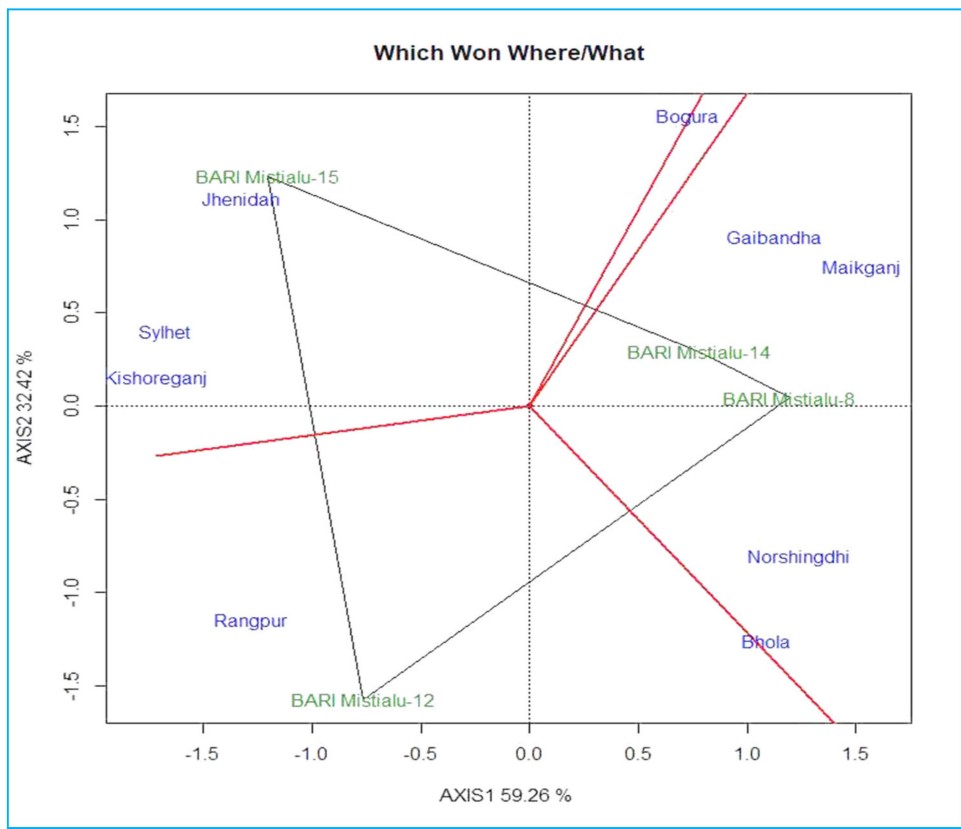

**Figure 5.** The polygon view of genotype and genotype x environment (GGE) biplot for the identification winning of sweet potato genotypes and related mega-environments.

### 3.5. Ranking of Genotypes

3.5.1. Ranking Genotypes Relative to the Perfect Genotypes

A perfect cultivar/genotype is characterized by a higher mean yield with higher constancy. The center of in concentric circles (Figure 6a) signifies the location of a perfect genotype, which is distinguished by a projection on the average-environment axis that is equivalent to the lengthiest vector of the genotypes that had exceeding-mean yield and by a zero projection on the perpendicular line (zero changeability across environments). A genotype is even more looked-for if it is nearer to the ideal genotype. While such an ideal genotype might not exist, it could be used as a condition for variety/genotype assessment [35]. This is due to the initial units of yield in the genotype-focused scaling of both PC1 and PC2 (Figure 6b), and the units should also be the original unit of yield under AEC abscissa (mean yield) and ordinate (stability). In addition, the distance unit between the ideal genotype and genotype, in turn, is the original unit of yield also. Thus, it is assumed that stability and mean yield are equally important based on the genotype-focused scaling ranking [19]. Therefore genotypes G4 ('BARI Mistialu-15'), which clearly fell into the center of concentric circles, were perfect genotypes in terms of good yield ability and stability, compared with the rest of the genotypes. Moreover, G3 and G1 may be viewed as advantageous genotypes, which are located on the next concentric circle.

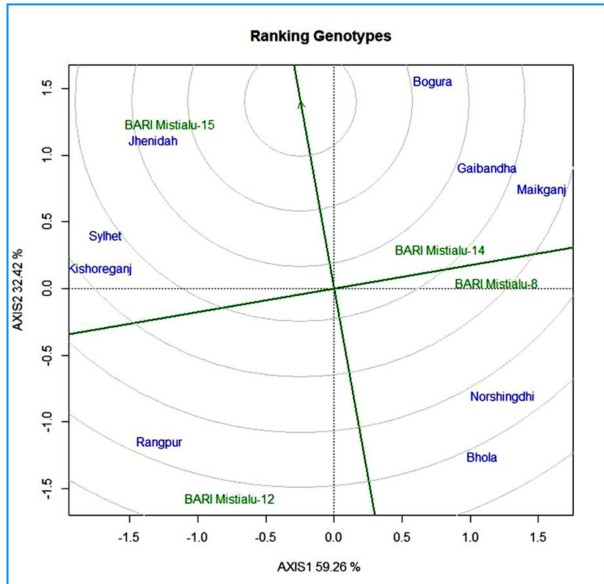 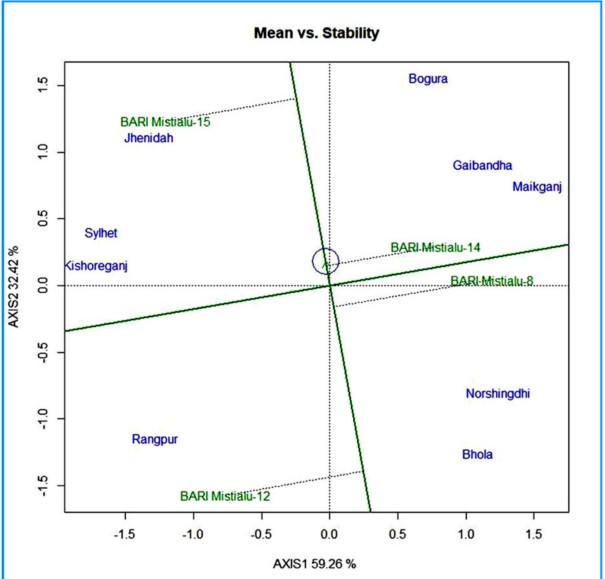

**Figure 6.** (**a**): GGE biplot showing the ranking of the sweet potato genotypes and related mega-environments and (**b**): GGE biplot showing the ranking of the sweet potato genotypes relation to mean vs. stability.

### 3.5.2. Ranking Environment Relative to the Perfect Environment

The GGE biplot way that is measuring representativeness is to describe a mean environment and practice it as a situation or benchmark. The small circle indicates the mean environment (Figure 7a). The perfect environment of the small circle with an arrow pointing to it is the most discerning of genotypes and yet representativeness of the other test's environments. Therefore, Bogura was the maximum wanted test environment, followed by Jhenaidah and Gaibandha. Figure 6 is the similar GGE biplot as (Figure 7b) suppose that it is based on environment-focused scaling [22]. This kind of AEC can be denoted as the "Discriminating power vs. Representativeness" view of the GGE biplot and can benefit the assessments of each of the test environments.

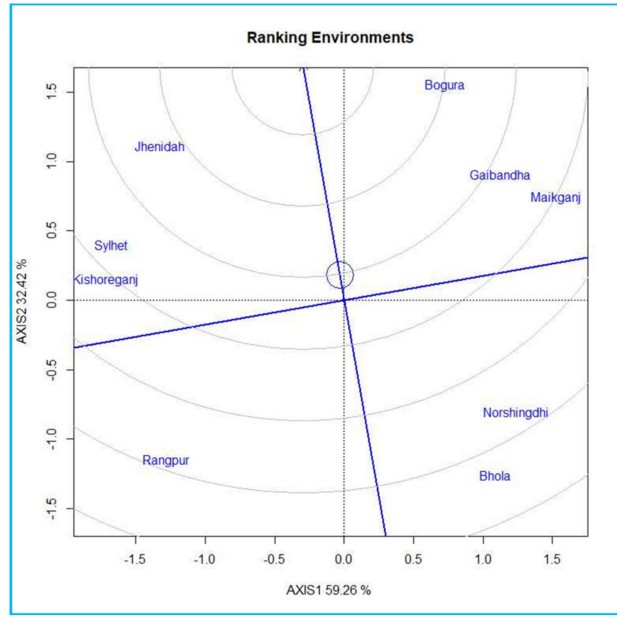 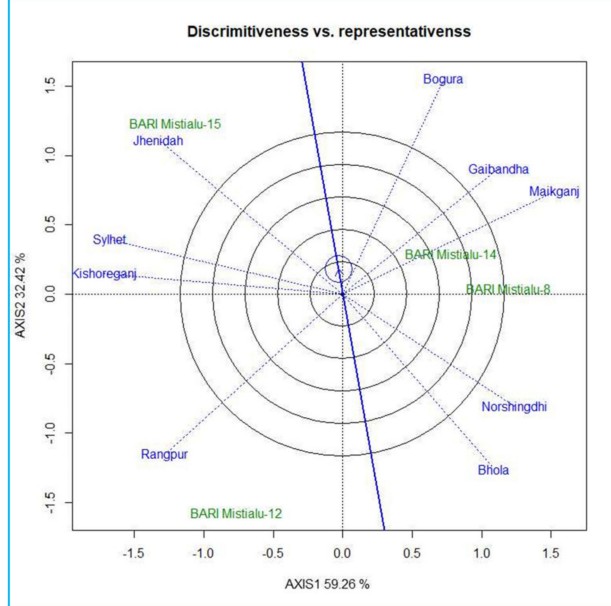

**Figure 7.** (**a**): GGE biplot based on environment-focused scaling for comparison the environments with the perfect environment and (**b**): GGE biplot based on discriminating power and representativeness of the test environment.

## 4. Conclusions

The tuberous root yield of BARI -released sweet potato varieties greatly influences the acceptance and adoption of varieties by farmers in different agro-ecological zones in Bangladesh. Among the four orange-fleshed sweet potato varieties evaluated across these locations, three varieties, namely G1 ('BARI Mistialu-8'), G2 ('BARI Mistialu-12') and G3 ('BARI Mistialu-14') were selected based on their better root yield, organoleptic evaluation and stability. These sweet potato varieties had wider adaptability and stability across the tested agro-ecological zones in Bangladesh. More importantly, these varieties were also selected by farmers as the best and ranked 1st, 2nd and 3rd among the tested varieties. Usually, the present experiment showed the opportunity of breeding sweet potato varieties for higher yield and wider adaptability throughout the country. Therefore, these sweet potato varieties may be presented for the nationwide cultivation that is currently grown in Bangladesh.

**Author Contributions:** Conceptualization, A.A.M., M.J.A., M.S.H.M., M.A.A., H.C.M., M.S.A., M.A.I., M.A.H.T., M.Z.F., M.R.A., M.F.H., M.M.A., M.S.I., and A.H.; methodology, A.A.M., M.J.A. software, M.A.A. and M.J.A.; validation, M.A.A. and M.J.A.; formal analysis, M.A.A., M.J.A. and A.H.; investigation, A.A.M. and M.J.A.; resources, A.A.M.; data curation, A.A.M., M.J.A. and A.H.; writing—original draft preparation, A.A.M., M.J.A., M.S.H.M., M.A.A., H.C.M., M.S.A., M.A.I., M.A.H.T., M.Z.F., M.R.A., M.F.H., M.M.A. and M.S.I.; writing—review and editing, M.M.H., E.S.D. and A.H.; visualization, A.A.M., and M.J.A.; supervision, A.A.M.; project administration A.A.M. and M.M.H.; funding acquisition, M.M.H., A.H. and E.S.D. All authors have read and agreed to publish the current version of the manuscript in *Sustainability*.

**Funding:** The current work was funded by the Director-General of Bangladesh Agricultural Research Institute (BARI), Joydebpur, Gazipur and Taif University Researchers Supporting Project number (TURSP 2020/85), Taif University, Taif, Saudi Arabia.

**Institutional Review Board Statement:** Not applicable.

**Informed Consent Statement:** Not applicable.

**Data Availability Statement:** Data recorded in the current study are available in all Tables and Figures of the manuscript.

**Acknowledgments:** We earnestly thank the Director-General of Bangladesh Agricultural Research Institute (BARI), Joydebpur, Gazipur for awarding research expenses and delivering the services for this study. The authors also extend their appreciation to Taif University for funding current work by Taif University Researchers Supporting Project number (TURSP 2020/85), Taif University, Taif, Saudi Arabia. We also thank to EHM Shofiur Rahaman, Senior Project Manager, International Potato Center (CIP), Dhaka 1230, Bangladesh for providing plant materials and resource for the research.

**Conflicts of Interest:** The authors declare no conflict of interest.

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
