# Peer review of "Farmers’ Preference, Yield, and GGE-Biplot Analysis-Based Evaluation of Four Sweet Potato (Ipomoea batatas L.) Varieties Grown in Multiple Environments"

_sustainability, doi:10.3390/su13073730_

Round 1

Reviewer 1 Report

After careful reading of the manuscript, this reviewer highlights a notable improvement in the writing of the manuscript and in the display of graphics. Since the purpose of the work is the evaluation of farmers' preferences and the organoleptic satisfaction of consumers on  four sweet potato (Ipomoea batatas L.) varieties,  while remaining an experiment with a low analytical claim, this reviewer considers the work as a whole acceptable.

Author Response

Response to Reviewer_1 comments

Reviewer-1

Reviewer comments

Response

After careful reading of the manuscript, this reviewer highlights a notable improvement in the writing of the manuscript and in the display of graphics. Since the purpose of the work is the evaluation of farmers' preferences and the organoleptic satisfaction of consumers on four sweet potato (Ipomoea batatas L.) varieties, while remaining an experiment with a low analytical claim, this reviewer considers the work as a whole acceptable.

Thanks for your valuable remarks for our manuscript which may be allowed to publish our findings in the journal.

Reviewer 2 Report

Dear Authors, some comments on the manuscript “Farmers’ preference, yield and GGE-biplot analysis-based evaluation of four sweet potato (Ipomoea batatas L.) varieties grown under multiple environments”

-Technical note - notation of citations in the text, data records, required formatting - different font, e.g. line 71, 130, 202,267,144, 376

- equals sign lines 212-220 - display badly

Paragraph 2.2 Experimental design and plant material

-No information on harvest dates in 2020

- Table 3 - no information on harvest dates in 2020

- Fig 1 and fig 2 - averages for years in given months or given months in which year, please specify in captions

-Table 4 - what years this average results - please specify in captions

Author Response

Response to reviewer_2 comments

Reviewer-2

Reviewer comments

Response

Technical note - notation of citations in the text, data records, required formatting - different font, e.g. line 71, 130, 202,267,144, 376

Thanks for your suggestions. We have thoroughly checked the whole manuscript and corrected where necessary

equals sign lines 212-220 - display badly

The suggestion has been checked and corrected

Paragraph 2.2 Experimental design and plant material

-No information on harvest dates in 2020

Table 3 - no information on harvest dates in 2020

As per your comments, the information on planting and harvesting dates have been added in Table 3

Fig 1 and fig 2 - averages for years in given months or given months in which year, please specify in captions

Information on crop season has been specified in the caption of Fig 1 and 2

Table 4 - what years this average results - please specify in captions

Information on crop season has been also specified in the caption of Table 4

Reviewer 3 Report

The manuscript “Farmers’ preference, yield and GGE-biplot analysis-based evaluation of four sweet potato (Ipomoea batatas L.) varieties grown under multiple environments” seems to be an exciting topic. The authors tried to explain environmental variability on yield and quality changes of some sweet potato cultivars and their possible interactions. Although it is challenging to correlate the yield and quality with environmental variations. They also tried to draw a relationship between environment and genotypes to find out the most suitable cultivars. However, some queries need to address to improve the manuscript. I suggest including some information for which I recommend for a minor revision.

Line 334 & 336: Please check the figure number mentioned here is correct or not.

Line 384: Please write the exact short name for each genotype

The results focused mainly on root yield and organoleptic test. It could have been better to incorporate some other field-oriented data like insect infestation, disease incidence, physiological disorders etc. as like the article “Islam, J.; Choi, S.P.; Azad, O.K.; Kim, J.W.; Lim, Y.-S. Evaluation of Tuber Yield and Marketable Quality of Newly Developed Thirty-Two Potato Varieties Grown in Three Different Ecological Zones in South Korea. Agriculture 202010, 327. https://doi.org/10.3390/agriculture10080327”

As the authors emphasized vitamins and mineral contents in sweet potato variety selection, it could better analyze the nutrients contents of all 4 genotypes.

Author Response

Response to Reviewer-3 comments

Comments and Suggestions for Authors

Reviewer Comments

Response

Reviewer-3

The manuscript “Farmers’ preference, yield and GGE-biplot analysis-based evaluation of four sweet potato (Ipomoea batatas L.) varieties grown under multiple environments” seems to be an exciting topic. The authors tried to explain environmental variability on yield and quality changes of some sweet potato cultivars and their possible interactions. Although it is challenging to correlate the yield and quality with environmental variations. They also tried to draw a relationship between environment and genotypes to find out the most suitable cultivars. However, some queries need to address to improve the manuscript. I suggest including some information for which I recommend for a minor revision.

Thanks for your suggestions. We have thoroughly checked the whole manuscript and corrected where necessary

Line 334 & 336: Please check the figure number mentioned here is correct or not.

In the paragraph, we try to incorporate the relationship between environment and genotypes

Line 384: Please write the exact short name for each genotype

As per the suggestion we have been used short name for all used varieties

The results focused mainly on root yield and organoleptic test. It could have been better to incorporate some other field-oriented data like insect infestation, disease incidence, physiological disorders etc. as like the article “Islam, J.; Choi, S.P.; Azad, O.K.; Kim, J.W.; Lim, Y.-S. Evaluation of Tuber Yield and Marketable Quality of Newly Developed Thirty-Two Potato Varieties Grown in Three Different Ecological Zones in South Korea. Agriculture 202010, 327. https://doi.org/10.3390/agriculture10080327”

We have been added the suggested reference

As the authors emphasized vitamins and mineral contents in sweet potato variety selection, it could better analyze the nutrients contents of all 4 genotypes.

Thanks for your good suggestion. Table 2 already have the information for all used varieties

Comments in attached PDF file

Besides these edits, we have been also thoroughly revised the whole manuscript as per request in the PDF file

This manuscript is a resubmission of an earlier submission. The following is a list of the peer review reports and author responses from that submission.

Round 1

Reviewer 1 Report

Manuscript sustainability-1082729 analyses the multi‐environment trial (MET) data of four new sweet potato varieties by using a genotype main effect (G) plus genotype‐by‐environment (GE) interaction (G+GE) biplot method. The manuscript fit into Sustainability aims and scope at the limit, being more suitable for other MDPI journals - e.g., Agronomy.

Manuscript needs improvements before publication. The title should be upgraded – farmers preference were established by organoleptic analysis. Therefore, both ”famers” preference” and “organoleptic” are redundant. My suggestion is to retain only “farmers preference” – this justify the manuscript submission to Sustainability.

Abstract is too long and  need to be shortened to maximum 200 words.

Introduction section, paragraph related to benefits of (G+GE) biplot analysis (L98-l122) must be re-written because it is not clear right now.

Material and Method section, subsection 2.1. Experimental locations, information regarding multi-year average temperature and rainfall for each location must be included.

Material and method Section, subsection 2.2 Experimental design and plant materials, the 34 complete block design (described in the Abstract) must be explained.

As a general rule, in a scientific paper, Figures shall stand alone from the main text.  This is not the case for Figure 4, Figure 5, Figure 6, and Figure 7. More explanation for understanding the figures must be included in the figure caption, to facilitate the understanding of the figures without referring to the main text.

Others required corrections are bellow.

L106, multi‐environment trial (MET)  must be here defined – it is used acronym MET in the rest of the manuscript.

L127- L128, “Upbringing these ideas in mind, the trials were commenced~. It is unnecessary and need to be deleted.

L155-L156. It must  be underlined that fertilization was done with composted “cow dung”. Un-composted cow dung is a source of water, soil and food contamination. In Bangladesh sanitary risk is mainly due to animal feces. -  Ercumen et al, 2017. Animal feces contribute to domestic fecal contamination: evidence from E. coli measured in water, hands, food, flies, and soil in Bangladesh. Environmental science & technology, 51(15), 8725-8734.

L228, Figure 3 must be cited in the main text.

L229, Figure 3 caption must include explanation regarding error bars standard deviation (SD) or standard error (SE)? Number of replicates must be also mentioned.

Author Response

Reviewer-1

Authors’ Response

The title should be

upgraded – farmers preference were established by organoleptic analysis. Therefore, both-”famers” preference” and “organoleptic” are redundant. My suggestion is to retain only “farmers reference” – this justify the manuscript submission to Sustainability.

Retain only Farmers’ preference as suggested by the reviewer

Abstract is too long and need to be shortened to maximum 200 words.

Abstract has been shortened within 200 words

Introduction section, paragraph related to benefits of (G+GE) biplot analysis (L98-l122) must be re-written because it is not clear right now.

This part has been rewritten as per your suggestion

Material and Method section, subsection 2.1. Experimental locations, information regarding multi-year average temperature and rainfall for each location must be included.

Average weather data (max and min temperature and rainfall) included in Fig. 1 and fig, 2

Material and method Section, subsection 2.2 Experimental design and plant materials, the 34 complete block design (described in the Abstract) must be explained.

Explained in the section 2.2 Experimental design and plant materials

As a general rule, in a scientific paper, Figures shall stand alone from the main text.  This is not the case for Figure 4, Figure 5, Figure 6, and Figure 7. More explanation for understanding the figures must be included in the figure caption, to facilitate the understanding of the figures without referring to the main text.

Thank you so much for your good suggestion. The caption of all figures has been revised

L106, multi‐environment trial (MET) must be here defined – it is used acronym MET in the rest of the manuscript.

Corrected in L260

L127- L128, “Upbringing these ideas in mind, the trials were commenced~. It is unnecessary and need to be deleted.

Upbringing these ideas in mind, the trials were commenced- deleted

L155-L156. It must  be underlined that fertilization was done with composted “cow dung”. Un-composted cow dung is a source of water, soil and food contamination. In Bangladesh sanitary risk is mainly due to animal feces. -  Ercumen et al, 2017. Animal feces contribute to domestic fecal contamination: evidence from E. coli measured in water, hands, food, flies, and soil in Bangladesh. Environmental science & technology, 51(15), 8725-8734.

Written well composted cowdung (it is not un-composed cowdung)

L228, Figure 3 must be cited in the main text.

Figure 3 already cited in the main text (sub-section 3.2)

L229, Figure 3 caption must include explanation regarding error bars standard deviation (SD) or standard error (SE)? Number of replicates must be also mentioned.

(mean ± standard error of the three replications)-added in the caption of Figure 3

Reviewer 2 Report

Dear authors,

This is an interesting and well written manuscript. I really enjoyed reading this research. However, I do have some concerns as reported below:

Line 162: “Data about the yield and yield supporting characters were recorded”. Which were these yield supporting characters that were recorded? Is there any other yield character apart yield on the data reported? I don’t think so. Please, precise these characters.

Line 229: “Figure 3. Yield performance of four sweet potato varieties in nine different locations”. Please, indicate what the meaning of bars is.

Line 238: “Genotype G3 238 (‘BARI Mistialu-14’) was the average 2nd highest root yielder”… you intend the 3rd not the 2nd. The 2nd was the Genotype G1. Please, check.

Line 252: An organoleptic evaluation was performed….Why no statistical test was performed for the organoleptic evaluation. How it’s possible to compare without a statistical analysis these results? Please, explain.

Moreover, I think that the discussion section is too short compared to the other sections.

Author Response

Reviewer-2

Authors’ Response

Line 162: “Data about the yield and yield supporting characters were recorded”. Which were these yield supporting characters that were recorded? Is there any other yield character apart yield on the data reported? I don’t think so. Please, precise these characters.

“Data about the yield and yield supporting characters were recorded “-deleted

Replaced by-“The tuberous root yield was collected from an area of 4 m2 (2 m X 2m) in each of the location and converted into t ha-1”

Line 229: “Figure 3. Yield performance of four sweet potato varieties in nine different locations”. Please, indicate what the meaning of bars is.

Added the meaning of bar by-“(mean ± standard error of the three replications)-added in the caption of figure 3”

Line 238: “Genotype G3 238 (‘BARI Mistialu-14’) was the average 2nd highest root yielder” you intend the 3rd not the 2nd. The 2nd was the Genotype G1. Please, check.

Actually G3 (BARI Mistialu-14) is the 3rd -ok and corrected in the main text

Line 252: An organoleptic evaluation was performed. Why no statistical test was performed for the organoleptic evaluation. How it’s possible to compare without a statistical analysis these results? Please, explain.

(mean ± standard error of the ten replications)-added in the caption of figure 4

Moreover, I think that the discussion section is too short compared to the other sections.

Added discussion in yield and organoleptic evaluation part.

Reviewer 3 Report

The authors, in this study, explore the ability of four sweet potato cultivars to respond in pedoclimatic adaptability, yield and product quality to 8 regions with different latitudes in  Bangladesh. They also evaluate the classification of genotypes with respect to perfect genotypes.

The objective of the research article is significant and has a strong application impact, both for farmers and consumers. Authors have judiciously gone through relevant literature before the planning of research, which is highly appreciable. The methodologies adopted for the investigations, unfortunately  are not entirely standard and appropriate for a rigorous scientific article. The PANEL test entrusted to only 10 people cannot replace with a simple sensory perception what the instrumental analysis of the product can indicate, including the fiber content. I remind the authors that their manuscript has been submitted to a scientific journal that is aimed at an audience of international scholars and that they use this analytical data to compare them with those in their possession for the genetic improvement and nutritional quality of the cultivars. The analytical sheet of the media used, even if not exhaustive, can be accepted, but the monuscript excluding the content of vitamin A, completely lacks a sheet of the nutritional values ​​of the genotypes used. Panel tests in scientific work are only useful as a complement to analytical and experimental data and employ a sample of professionally trained individuals with a strong sense of taste and smell.

Therefore I advise the authors to perform nutritional analyzes on extracts of the 4 genotypes as follows: Organic compounds - Carbohydrates, proteins, fibers, sugars and starch. Minerals: Anything you can determine, especially K, Zn, Fe, Mg and Se.Vitamins: Anything you can determine, especially Ascorbic Acid (VItamin C), Thiamine (Vitamin B 1), Niacin (Vitamin B3)and Pyridoxine (Vitamin B6). Finally in Materials and Methods:  Figure 2 the (y) axis of temperatures does not appear. in addition, it is useful to indicate what the error bars refer to.

Author Response

Reviewer-3 Comments

Authors’ response

The authors, in this study, explore the ability of four sweet potato cultivars to respond in pedoclimatic adaptability, yield and product quality to 8 regions with different latitudes in Bangladesh. They also evaluate the classification of genotypes with respect to perfect genotypes.

The objective of the research article is significant and has a strong application impact, both for farmers and consumers. Authors have judiciously gone through relevant literature before the planning of research, which is highly appreciable. The methodologies adopted for the investigations, unfortunately are not entirely standard and appropriate for a rigorous scientific article. The PANEL test entrusted to only 10 people cannot replace with a simple sensory perception what the instrumental analysis of the product can indicate, including the fiber content. I remind the authors that their manuscript has been submitted to a scientific journal that is aimed at an audience of international scholars and that they use this analytical data to compare them with those in their possession for the genetic improvement and nutritional quality of the cultivars. The analytical sheet of the media used, even if not exhaustive, can be accepted, but the manuscript excluding the content of vitamin A, completely lacks a sheet of the nutritional values ​​of the genotypes used. Panel tests in scientific work are only useful as a complement to analytical and experimental data and employ a sample of professionally trained individuals with a strong sense of taste and smell.

Therefore I advise the authors to perform nutritional analyzes on extracts of the 4 genotypes as follows: Organic compounds - Carbohydrates, proteins, fibers, sugars and starch. Minerals: Anything you can determine, especially K, Zn, Fe, Mg and Se.Vitamins: Anything you can determine, especially Ascorbic Acid (VItamin C), Thiamine (Vitamin B 1), Niacin (Vitamin B3)and Pyridoxine (Vitamin B6). Finally in Materials and Methods:  Figure 2 the (y) axis of temperatures does not appear. in addition, it is useful to indicate what the error bars refer to.

Thank you so much for your good suggestion. As per your suggestion Materials and Methods of the article has been reorganized for better understanding.

We are sorry that we were unable to do your suggested analysis for quality parameters. But as it is agronomic research, we provided major characteristics of all varieties in Table 2.

As per your suggestion, Fig. 2 has been revised for showing temperatures clearly.

Reviewer 4 Report

Dear Authors,

The paper is an interesting case study. Presented paper for review covers a wide range of research, creating one whole, from morphological and organoleptic tests to consumer evaluation. In a fair way, all data were presented; which, sweet potato varieties yield the best and why. The paper deals with the regionalization of cultivation of potato varieties and consumer evaluation- showing in which regions of the country there will be demand for given varieties.

The presented work is also supported by statistical analyzes, however, are the annual data are sufficiently representative?

Comments

  1. In my opinion, despite the large contribution to the analyzes and the wide scope of the research, they are not statistically representative (it is only one year of observation). Is there any further research going on? It would be good to include the results from the following years of observation. Because at the moment these are the results of preliminary research. At the moment I am waiting for information about further years of observations, that is why the status of the article - major revision. However, if it is only one-year study, I will apply for reject of paper in the next step.
  2. Please read the MS once more and correct any minor shortcomings, e.g. punctuation, etc.

Author Response

Reviewer-4

Authors’ Response

  1. In my opinion, despite the large contribution to the analyzes and the wide scope of the research, they are not statistically representative (it is only one year of observation). Is there any further research going on? It would be good to include the results from the following years of observation. Because at the moment these are the results of preliminary research. At the moment I am waiting for information about further years of observations, that is why the status of the article - major revision. However, if it is only one-year study, I will apply for reject of paper in the next step.

Although it is one year information, but the research was conducted in multiple locations. This study is one type of a model for understanding the farmers’ preference, yield and GGE-biplot analysis-based evaluation of crop variety (s) in multiple environments.

  1. Please read the MS once more and correct any minor shortcomings, e.g. punctuation, etc.

Corrected the minor shortcomings

Round 2

Reviewer 3 Report

I appreciate the effort of the authors, and I recognize that the work is acceptable, but does not fall within the parameters of this journal.

Reviewer 4 Report

The authors referred to the alleged shortcomings.

The authors made corrections regarding abbreviations, punctuation, etc. as recommended,

The above research, despite the wide spectrum of analyzes, in my opinion should be repeated -years of research.